# Enabling Fog–Blockchain Computing for Autonomous-Vehicle-Parking System: A Solution to Reinforce IoT–Cloud Platform for Future Smart Parking

**DOI:** 10.3390/s22134849

**Published:** 2022-06-27

**Authors:** Aamir Shahzad, Abdelouahed Gherbi, Kaiwen Zhang

**Affiliations:** Department of Software and IT Engineering, École de Technologie Supérieure, Montréal, QC H3C 1K3, Canada; abdelouahed.gherbi@etsmtl.ca (A.G.); kaiwen.zhang@etsmtl.ca (K.Z.)

**Keywords:** Internet of Things, cloud computing, fog/edge computing, smart autonomous parking system, radio-frequency identification, blockchain, cryptography

## Abstract

With the advent of modern technologies, including the IoT and blockchain, smart-parking (SP) systems are becoming smarter and smarter. Similar to other automated systems, and particularly those that require automation or minimal interaction with humans, the SP system is heuristic in delivering performances, such as throughput in terms of latency, efficiency, privacy, and security, and it is considered a long-term cost-effective solution. This study looks ahead to future trends and developments in SP systems and presents an inclusive, long-term, effective, and well-performing smart autonomous vehicle parking (SAVP) system that explores and employs the emerging fog-computing and blockchain technologies as robust solutions to strengthen the existing collaborative IoT–cloud platform to build and manage SP systems for autonomous vehicles (AVs). In other words, the proposed SAVP system offers a smart-parking solution, both indoors and outdoors, and mainly for AVs looking for vacant parking, wherein the fog nodes act as a middleware layer that provides various parking operations closer to IoT-enabled edge devices. To address the challenges of privacy and security, a lightweight integrated blockchain and cryptography (LIBC) module is deployed, which is functional at each fog node, to authorize and grant access to the AVs in every phase of parking (e.g., from the parking entrance to the parking slot to the parking exit). A proof-of-concept implementation was conducted, wherein the overall computed results, such as the average response time, efficiency, privacy, and security, were examined as highly efficient to enable a proven SAVP system. This study also examined an innovative pace, with careful considerations to combatting the existing SP-system challenges and, therefore, to building and managing future scalable SP systems.

## 1. Introduction

The Internet of Things (IoT) empowers smart connectivity and advanced communication paradigms in wireless sensor networks (WSNs) to, among other things, enable well-performing intelligent transportation systems (ITSs), including SP systems, which are considered integral parts of innovative smart-city projects [1,2]. The IoT is a generic platform that is used to overcome the technological limitations of existing transportation systems, such as SP systems, traffic-congestion systems, and energy-consumption systems; thereby, IoT-enabled features, such as monitoring, controlling, and data analytics, can strengthen the performances of larger SP systems in metropolitan areas [3]. For example, the SP system can play an essential role in saving drivers time when searching for parking in urban areas on demand; this can be of great assistance to minimizing traffic congestion and saving energy (e.g., gasoline), which further leads to a reduction in air pollution. As a part of the SP system, the parking payment as per the occupation of the driver can easily be made, with some built-in payment methods online, which therefore saves the driver time (instead of waiting in a long row), without interrupting other drivers [1,4].

The technological evolution of WSNs makes it viable to use various types of sensors, such as ultrasonic, magnetic, radio-frequency identification (RFID), optical, infrared (IR), Honeywell, etc., as well as communication protocols such as Bluetooth, Wi-Fi, ZigBee, 6LoWPAN, etc., to acquire the vacant parking information for the end user that is mainly connected through cellular applications and functions via backend servers (e.g., cloud servers) [5,6]. Thus, the usage of various sensors, embedded technologies, and their deployments in WSNs play vital roles in fulfilling the technological demands of modern SP systems [2,7]. However, such sensors and wireless communication standards come with several limitations to their designs, operational incompatibilities, protocol interoperability, data exchanging, environmental obstacles, etc., that disable the quality of service (QoS) (for example, in the case of managing and controlling a large SP system that offers a high probability of routing the overall parking information to its end user on demand in real time) [8,9]. In practice, the SP system is operational in WSNs, wherein field sensors and devices are networked to exchange information with a central controller over the Internet. The gateways are used to collect information from the field sensors or devices to transmit the carried information back to the central controller, or cloud server. The cloud server is designed to collect, store, and analyze large amounts of data from several distributive parking stations, and therefore manages the enormous end-user requests for parking. However, the cloud-centric platform has limitations in fulfilling the vast growing demands of modern SP systems (for example, an SP system that has connectivity with several parking stations, and each station is composed of several thousand parking slots) [1,10,11].

Instead of having such a centric nature, with its constraints, the cloud-computing platform offers several features and services to manage the operational activities of numerous SP systems (detailed literature is provided in [12,13,14,15,16,17,18,19,20]). Traditionally, the IoT also uses a cloud platform to manage its vast data storage, management, analysis, and even the users’ query management via cellular applications that run through the cloud [16,17,18]. Thus, together, the IoT–cloud platform can offer various solutions to build an effective SP system [16]. However, similar to the cloud, the IoT is entirely a centric and resource-constrained platform that thus lacks the capabilities for managing large SP systems. A single point of failure (SPOF) caused during data storage, periodic maintenance, software updates and scheduling, excessive traffic, etc., can lead to the interruption of the entire central system, which therefore influences the probability of service on time; for example, in the SP system, the accuracy of obtaining an indication as to vacant/nonvacant parking information within a certain period can suffer [19,20,21]. Therefore, for an IoT–cloud-centric platform, these are the main challenges, which also include the issue of centralized security management, which can be tackled through the incorporation of emerging technologies, such as fog computing and blockchain [9,22,23].

Fog computing is not only a solution to bring and manage the storage and computation closer to the edge devices (for example, in a distributive IoT–cloud environment), but it is also an integral shift in support of various services, such as virtualization, remote access, mobility, and cellular applications, which boost the performance of the existing central cloud by offering minimal latency and high throughput, and especially for the resource-constrained IoT platform [10]. However, fog computing is still not a mature technology, such as the IoT and cloud, as this technology has several challenges related to development costs, performance gains, computation complicities, QoS, scalability, and traffic management, to enable an effective fog infrastructure for systems such as SP [10,24]. Undoubtedly, incorporating fog computing can strengthen the capabilities of the IoT–cloud platform and therefore manage a scalable well-performing SP system [1,10]; however, enabling secure end-to-end fog communication is a challenging issue, as this technology lacks the design to combat cybersecurity issues [25]. To address this, blockchain is an emerging solution to mitigate the cybersecurity, security, and privacy issues, as blockchain offers a secure ledger solution to keep a record of every transaction, and it employs resilient cryptographic methods, such as hashing for data integrity, public-key cryptography, and nonbreakable digital signatures. Moreover, blockchain empowers fine-grained transparency to deliver and ensure an end-to-end information stream (for example, the successful transactions between edge devices to fog to cloud to end users, and vice versa) [11,26]. In particular, the major limitations of blockchain, such as mining, distributive data storage, computation complexities, and energy consumption, can be addressed through the incorporation of fog computing on massive IoT platforms [26].

In this modern technology era, most vehicular companies, such as Toyota, Telsa, Audi, Volvo, Mercedes-Benz, Nissan, Bosch, etc., have started to develop AVs, as reported in [27,28]. Figure 1 illustrates that the successful navigation of AVs on the road can be possible by employing a range of technologies, including a range of sensors, cameras, the global positioning system (GPS), a highly efficient radar system, high-definition radio antennas, and connectivity to a faster broadband network, such as 5G [29]. Nowadays, an advanced driver assistance system (ADAS) is a great choice for vehicular companies to use in vehicles to perform essential safety features while driving, such as automatic braking, the provision of blind-spot information, collision detection, lane warning, steering support, cruise control, and others; in addition, autonomous vehicular technology (AVT) brings the ADAS to the next level, where no physical drivers are operating the vehicles [29,30]. Essentially, for vehicle identification and location tracking, the usage of RFID tags and sensors, including GPS sensors, has always been of great importance to identifying real vehicles and tracking their locations. The collective information is usually shared with security agencies, such as traffic police and insurance companies, to identify the vehicle and its location in case, for example, the vehicle is stolen. In short, all over the world, sensing technology and its numerous applications have been employed in vehicles for various purposes. Sensors are used in vehicles and are networked to deliver collected information back to the cloud, or other vehicular-based backend servers. However, building and managing such a crowded vehicular or transportation system (for example, a large-scale IoT-enabled SP system) through a centric platform such as the cloud is not a long-term effective solution to address the limitations of computation, storage, and cybersecurity [21,22,23,24]. For that, the introduction of the emerging fog and blockchain technologies can be an inclusive solution to overcoming the challenges of IoT–cloud-centric platforms to build smart transportation and to manage its current and future demands (for example, SP systems) [1,21].

This study provides inspiration for building and managing future SP systems, and specifically for accommodating AVs searching for parking, both indoors and outdoors, in metropolitan areas. For this, the proposed SAVP system employs fog computing and a lightweight integrated blockchain and cryptography (LIBC) module as add-ons to the existing IoT–cloud platform, thereby enabling a scalable SP system. The SAVP system is not only just a solution to designing and modeling a scalable SP system, but it is also a solution that pays careful attention to leveraging the optimal performances, and mainly in terms of computational latency reduction, an improved query-response time, location awareness, and self-checking security and privacy. The supportive location-awareness solution of the SAVP system can track AVs where the GPS is not operational (for example, in the case of indoor parking). The main objectives of this study that actively contributed to building and managing an effective and well-performing SAVP system are as follows:The design of a network-friendly architecture, in which fog and blockchain technologies are employed in the existing IoT–cloud platform to fulfill the requirements of the proposed system;The management of traffic in a distributive IoT–cloud platform, where a number of parking slots are functional with RFID readers of the active type, and a fog node is employed in each parking station. A fog node is an intermediate controller to manage overall parking operations, such as massive transactions from edge nodes, the LIBC security module, the blockchain module, etc;To speed up the acquisition at each fog node, the LIBC module uses Advanced Encryption Standard (AES) 256-bit and SHA-256-bit algorithms to perform the security checks, such as the authentication and verification of the AV;To ensure QoS at each stage of the SAVP system, four communication scenarios were defined: edge-to-fog, fog-to-cloud, fog-to-fog, and cloud-to-cloud, where every carried transaction is recorded onto the blockchain. To manage privacy, each node has an anonymous identity for communication;To demonstrate the existence of the study, a proof-of-concept implementation was conducted, and a theoretical analysis was performed to evaluate the strength of the SAVP system.

The rest of the paper is organized as follows. An in-depth literature review is conducted in Section 2. Section 3 details a system’s architecture, where a three-layer-system architecture is detailed to manage both indoor and outdoor parking. Section 4 details the proposed SAVP-system model, where several participating entities are defined, and communication scenarios are highlighted to carry the information between the system’s entities. We present the proof-of-concept implementation that was conducted, the results that were measured, and the relative discussions that were had, together with theoretical evidence, in Section 5. Section 6 highlights the main challenges of the SAVP system, and some interesting future directions to combat these challenges. Section 7 concludes the proposed overall study and its main goals.

## 2. Literature

Conventionally, both the IoT and the cloud have been considered integrated platforms, as the IoT started becoming functional with the collaboration of the cloud-computing platform, which is a client-server architecture that delivers services and operations to various systems, including SP systems [17,21]. The SP system uses the IoT–cloud-network architecture to enable the visibility of its nodes over the Internet to share information with the cloud controller, which enables a bidirectional link to execute various parking operations [31,32]. To do so, various sensors (and devices), such as ultrasonic, magnetic, RFID, etc., are used to find vacant parking (e.g., indoor/outdoor parking); together, RFID and Bluetooth communication standards are used to locate the available parking location, as well as the location of the vehicle [33,34]. Depending on the requirements, IoT-enabled three- or five-layer-network architectures can be used to perform various tasks in SP systems [31,32,33]. In the IoT, the three-layer architecture is comprised of: (1) The perception layer, which is a bottom layer that is composed of various types of sensors, devices, or hardware equipment to enable connectivity with the physical environment. (2) The network layer, which stands above the perception layer. As the name suggests, this layer is responsible for performing networking services and operations through connectivity with various data-transmission or routing protocols and standards. (3) The application layer, which is the uppermost layer and is responsible for managing the traffic from the network layer and end-user requests for services [35]. Figure 2 illustrates the IoT five-layer architecture, where there are an additional two layers that are not available in the IoT three-layer architecture, which are: (1) The middleware layer, which stands between the application and network layers and is responsible for enabling the interactions among devices at the perception layer, and for managing data using various programming models and logics. It is a layer that uses various approaches to enable communication among those objects and modules that are not otherwise functional. (2) The business layer, which stands on top of the application layer and is responsible for managing various user-defined business models, applications, and even users’ privacy [31,32,33,34,35].

Kamran et al. [11] examined the common underlying limitations, such as latency and high-bandwidth costs, of cloud-based developments for the existing SP systems, and then proposed a fog-based parking solution as an add-on to minimize the bandwidth costs and increase the average response time. A parking system, as a proof-of-concept implementation, was simulated using the iFogSim simulator, and the measured performances were compared with the existing available cloud-based parking solutions. The network architecture of the parking system (i.e., one parking area used a single fog node, and multiple parking areas employed multiple fog nodes) is composed of three layers: (1) a device layer, which comprises cameras that are installed and used to determine the availability of parking slots for end users; (2) the second layer, or fog layer, which connects to the cameras via microprocessors; (3) the third layer, or cloud layer, where the fog nodes are connected to the cloud server via a proxy server, which is responsible for storing and managing massive amounts of data that are continuously received from fog nodes for long-term storage.

In [17], the authors proposed an algorithm that computes a function (F(α, β)) where α and β are coefficients that present the distances between the parking stations and available parking slots, which thereby enhanced the performances of existing cloud-based SP systems. The performance metrics (e.g., time and distance) are considered to compute the lowest cost to offer the better sites to find available parking in a shorter period of time. Moreover, a cellular application that runs on the android platform is used to search and then reserve the available parking slot. For that, an IoT prototype using Arduino Uno R3 and RIFD is implemented to detect the parking spaces, which are available close to the position of the end user [17,36]. Moreover, in [5], an IoT-enabled SP system was simulated using Arduino Uno and Raspberry Pi3 to manage automated multilevel parking architectures, wherein IR sensors are deployed to identify the free parking and are controlled by Arduino Uno, which is installed at each level, and all the parking levels are administered through Raspberry Pi3, which is located on the parking ground level.

O. Orrie et al. [36] proposed a wireless parking system that employed wireless sensors to obtain an indication of the vacant parking spots in a parking place. For that, a prototype was built using Raspberry Pi to retrieve parking information from field sensors that were connected with Arduino Uno comprised of a sub-1GHz CC1101 transceiver and ATmega 328 microcontrollers. The sensors were activated to enable communication through Wi-Fi, which thereby transmitted the parking information to the backend server, which was connected via the “TRENDnet 150 Mbps Wireless N USB Adapter”. The authors also used a cellular android application, set up with Google Maps, to search for a vacant parking spot; thereby, a user can obtain an indication of a closer parking spot within 2 km. In [37], the authors used a computer vision scheme (i.e., cloud vision API) to identify the car in the parking area by capturing and processing its number-plate images through the Raspberry Pi Camera Module, which has compatibilities with the cloud. Overall, the work performed was considered a fine solution to finding parking slots and ensuring parking security.

Lee et al. [33] set up a parking system that comprised an ultrasonic sensor for indoor parking and a magnetic sensor for outdoor parking to detect and locate vacant parking spots. Each parking field was installed with a sensor mote and a powerful microcontroller called Atmel ATmega128 MCU. Thus, the data collected from the mount sensor, including the sensor ID and USIM (Universal Subscriber Identity Module) ID, were always shared with a controller. A Bluetooth communication module was used to enable communication between each sensor node (vacant) and the user’s smart cellular phone. In [38], the authors considered the Travelling Officer Problem and formulated a system with the probability-based model (i.e., a spatial-temporal model) to minimize physical patrolling and to make the processing of infringement notices in a shorter amount of time cost efficient. A dataset or spatial-temporal data were collected from the existing on-street embedded sensors and were processed with the greedy and ant-colony-optimization (ACO) algorithms, which are considered parts of the probability model, for optimal performance estimation and pathfinding. The authors concluded that the conducted work outperformed the parking patrolling and maximized the infringement parking notices.

Cybersecurity is not the main concern of either IoT or cloud platforms (for example, to authenticate the real identities of participating nodes during data exchange and ensure data integrity in bidirectional communication) [39,40]. In [41,42,43,44], detailed literature is available to tackle the issues of security and privacy, in which various solutions have been employed, such as mutual authentication, the OAuth protocol, encryption, blockchain, etc., for numerous applications and systems that function under the IoT and cloud-computing paradigms. Considering the massive IoT platform, where roughly millions of devices are geographically distributed and connected to a cloud server, it is a challenging issue to address cybersecurity on such a heterogeneous platform, where field devices from distinct manufacturers, incompatible communication protocols, data standards, etc., are largely grouped [1]. In [11], IoT-enabled RFID technology was used to offer smart-object identification and tracking where data analytics and storage services are performed over the cloud. The distributed untrustworthy nature of the IoT and its connectivity to the distantly located cloud controller was considered to upgrade the overall system performance, and mainly the security at different levels of communication [45,46]. In [47], a mutual authentication protocol was used to offer a secure way to enable communication between the RFID tag and the reader, and the reader had a secure communication channel to obtain a response from the database server. Encryption schemes are highly proven solutions to build security for various applications; however, due to their computation complexities, these solutions have not been considered appropriable for RFID systems [48]. In [45], an authentication protocol is proposed to enable the end-to-end secure routing of information over communication channels, where a shared secret key is used by the system entities, from the tag to the reader to the cloud server, in an RFID system.

In [22], the IoT was considered as a cornerstone to address the goals of a circular economy (an economy model that encourages the reusability of the asset rather than waste), and it enabled fine-grained and uninterrupted object tracking. Authors have studied the IoT challenges of security and privacy and have concluded that a large number of devices generate and publish information on IoT platforms, without having proper mechanisms to address the information leakage over the Internet. To address these issues, edge and blockchain technologies are used together as a solution on large-scale and highly constrained IoT platforms. To perform the validation, or proof of work, IoT smart devices of type C are used to perform the mining process; however, the selection of miners that are authorized to perform the mining is not mentioned, wherein a new block is generated and off-loaded to the ledger available onto the edge nodes; this work, therefore, eliminated the limitation of the storage of each smart device and overcame the latency overhead that is required to publish a transaction or to add a new block to a distributive blockchain ledger. Furthermore, in [49], the authors propose a memory-efficient edge-computing solution to make possible the mobile blockchain, wherein mobile devices are used as miners’ nodes to perform the proof of work, and the edge nodes are considered as service providers to enable resources, such as computing and storage. The new blocks are generated and recorded onto the ledger available over the edge nodes rather than the resource-constrained mobile nodes. Particularly, edge nodes act as the blockchain service providers to enable resources to perform the PoW, and then record blocks onto a ledger in a peer-to-peer network. Alternatively, enabling a private blockchain (i.e., having a fixed number of trusted miners) is a solution to mitigate the cybersecurity issues of IoT and cloud platforms [11,50]. In [51], a DualFog–IoT system was modeled to address the IoT–fog integration with blockchain, in which the fog layer performs the computation and services virtually in two modes: (1) the fog cloud cluster, where it follows the conventional architecture of the IoT–cloud system; (2) the fog-mining cluster, where fog nodes are considered trust nodes to compute the validation or mining operation of the application running on the blockchain. The computed results, such as the reduction in the drop rate and minimization of the latency, energy resources, and blockchain-yielded security and privacy, fared comparatively better than the available central IoT platform.

Blockchain can play an important role in reducing the security and privacy barriers of IoT and cloud platforms; however, blockchain solutions have always been considered costly (e.g., performance costs) in terms of decentralization, computation, bandwidth overhead, and latency, which has enabled the compatible development of central real-time IoT platforms [11,22]. In [23], a lightweight blockchain approach was used to store and maintain transactions from the IoT network. For that, an immutable ledger was centrally deployed (i.e., cloud storage), which thereby reduced the overhead and energy consumption required in conventional blockchain distributive ledgers, to make the IoT platform resilient against security and privacy threats [22,26]. A broader concept of smart-home use was considered and evaluated to underpin the routing packets and computing overhead in IoT systems. However, the blockchain solution has limitations in providing efficient data access in real time, and the delay from the cloud service is estimated while responding to the end-user (e.g., smart vehicle) request [11,22,23]. In [52], a blockchain-enabled key management and distribution architecture is proposed as an alternative to third-party or central-authority involvement, and the designed architecture is built with fog nodes to reduce the latency and computation costs, and to enable privacy standards for control access in a hierarchical IoT platform. In [53], a distributed cloud system is proposed that uses blockchain standards and software-defined networking capabilities to enable fog nodes to be operational at the edge of the IoT network. Overall, the system model actively contributed to reducing the operational costs and latency, securing data access, and yielding on-demand access to cross-domain infrastructures in IoT platforms. In [42], blockchain Ethereum smart contracts were executed for authentic users to access devices, and the devices had connectivity to fog nodes that provided interaction with smart contracts in an IoT system. The outcomes of the proposed decentralization scheme, comprised of five main entities (the administrator node, intermediate fog nodes, cloud-computing server, end user, and field devices) outperformed the computation over the fog nodes and also the user’s authentication to communicate with various devices on a scalable IoT platform.

## 3. System Architecture

Similar to other sectors, IoT- and cloud-based solutions have also been largely deployed for transportation systems, including SP systems, which has thereby enabled smart connectivity and the management of numerous resources, data storage, and computation [1,9,13]. For a massive centric IoT–cloud platform, fog computing is an effective solution to address the limitations of computational latency and throughput, and blockchain is a proven solution to combat the notable security challenges, such as, but not limited to, authentication and data integrity [8,24]. In this study, we examined, in depth, the main limitations (i.e., computational latency, efficiency, privacy, and security) of the existing IoT–cloud-based developments for SP systems [11,12,13,14,15,16,17], and then proposed an SAVP system that deploys fog nodes as an extension to the centric cloud-computing platform that functions with blockchain capabilities to circulate and manage a massive amount of vehicular parking data in an IoT network to achieve better overall system latency, throughput, privacy, and security. The SAVP system’s design was limited or mainly targeted to accommodate AVs that have real identities (e.g., via RFID active tags) and that are looking for parking solutions. For security purposes, and specifically to manage the traffic in a massive IoT–cloud platform, the parking facility is only available to AVs that have already been registered under the SAVP system.

In reality, there are several companies that may, under the regulation of government licenses, issue vehicular tags of different types to be further installed by vehicular companies or third-party installation [27,28]. To build a future scalable SP system where all parking stations, both indoors and outdoors, will be autonomous and functional to accommodate a large number of vehicles, and especially self-driving vehicles, these identifications via tags are registered under government regulation, which thereby makes this a resilient solution to enable a secure SP system for vehicular parking in situations of, for example, self-driving cars. Moreover, this solution can identify the vehicle (e.g., whether an authorized vehicle to the system or not) and will accommodate parking both indoors and outdoors. The identification of a vehicle via a RFID tag, whether active or passive, has been a widespread solution; however, there is the major issue, among others, that a tag embedded within a vehicle, for example, can be taken off and/or stolen; therefore, a solution to this problem is to install several tags that have unique identifications for the different parts of the vehicle body. However, having a number of active tags embedded within the vehicle will be an expensive solution, whereas having passive tags could be a better solution to overcome the costs. In this study, we used active tags instead of passive ones because the active tags can fulfill the study’s requirements, as these tags have enough power and memory to store the security keys in order to perform one-way communication and authentication in the SAVP system.

Figure 3 illustrates the architecture of the SAVP system, which comprises three generic layers: (1) the perception layer; (2) the intermediate fog layer; (3) the cloud layer. This three-layer-system architecture is employed to manage both indoor parking, for example, in a multilevel building (MLB), and outdoor parking, which includes on-road parking (ORP) and open-premise parking (OPP). To facilitate a reliable parking solution, this study targeted the minimal utilization of the overall system resources, including the minimum latency and network bandwidth, to accommodate a large number of AVs that are searching for parking closer to their intended locations. For parking requests, the SAVP system deploys a reliable way to accommodate the numerous AVs in a minimal period of time, as the vehicles are fully autonomous; therefore, each AV is supposed to be programmed with a software-based solution in order to process the requests for parking. Thereby, upon successful confirmation of parking in the SAVP system, fog nodes will be active to identify the AV so that it can obtain access to the designated parking station, from the parking main entrance to the designated parking slot. At each fog node, the LIBC module functions to authenticate the AV before permitting parking access, and to record every transaction, whether successful or not, onto the blockchain. The details on each layer and its components that are employed in the proposed system architecture are described below.

### 3.1. Perception Layer

This layer is comprised of field devices that are resource-constrained in terms of computation, storage, and power; therefore, these devices are not capable of performing heavily complex computation and massive data storage (for example, the blockchain operations, such as proof of work and keeping a copy of the ledger, which is usually undefined in size, in the IoT platform). In our case, we employed IoT-enabled RFID devices of the active type, which are also resource-constrained devices. Particularly, each device’s internal storage was sufficient to store a unique identification (ID), a session key, and a hash of IDs in order to initiate communication with an IoT-enabled RFID reader. Each parking slot was installed with an IoT-enabled RFID reader and was able to obtain the reading from the RFID tag embedded in the AV. The number of parking slots in parking stations is entirely dependent on the area it occupies. In the situation of on-road parking, the RFID readers are mounted safely, with a parking panel closer to the edge of the road on the pavement. For both indoor and outdoor parking, their locations are fixed, and each parking slot is active with an IoT-enabled RFID reader via Ethernet to communicate with fog nodes using MQTT (Message Queue Telemetry Transport)**.** In the case of wireless connectivity to fog nodes in WLAN or WPAN, the parking slots are operational using Wi-Fi or ZigBee.

### 3.2. Intermediate Fog Layer

The fog layer functions as a high computing layer to perform various tasks at the edge of the SAVP system, which mainly include: keeping a record of the field devices (e.g., IoT-enabled RFID readers and tags), along with their unique IDs and hashes of IDs; the LIBC module to authentic the AVs at each stage of parking, from the parking entrance to the parking slot to the parking exit, and to continuously perform checks to record information on the parking activities; ensuring coded RFID-tag-data integrity, processing information back to the cloud for further analytics if required; long-term storage; data validation, etc. In short, in the SAVP system, each fog node acts as an intermediate node between the cloud and the edge nodes, or as a local administrator in the local area network (LAN), to manage the overall operations of the centric cloud closer to the edge nodes. Fog nodes are connected to the cloud through the gateways in a wide area network (WAN) via the global system for mobile communications (GSM) or long-term evolution (LTE).

### 3.3. Cloud Layer

In this layer, each cloud controller is superior and functions to provide various services through the fog nodes, which act as a platform-as-a-service in the SAVP system. The main goals of the cloud layer are: to manage end-user or AV requests for parking and massive amounts of traffic data from the fog nodes; to monitor the fog operational activities; to continuously update the repository with information from the fog nodes; to offer long-term data storage, including a hash of transactions; to perform analytics on the massive amount of data that are continuously or from time to time collected from the fog nodes. In this layer, services such as the certificate authority (CA), key distribution center (KDC), and validation are also performed. 

## 4. System Model

In the SAVP system, there are n parking stations ${\left(PS\right)}_{n}$, where n is some fixed value, and PS denotes a parking station, which is located in various locations in a metropolitan city (e.g., we consider the city of Montreal), such that:(1)$${\left(PS\right)}_{n}=\left\{{\left(PS\right)}_{1},{\left(PS\right)}_{2},\ldots \ldots ,\;{\left(PS\right)}_{n-1},{\left(PS\right)}_{n}\right\}$$

As mentioned, the SAVP system is comprised of three types of parking: MLB, ORP, and OPP, which are the most common types of parking that are mainly offered in metropolitan areas. For simplicity, we assumed the following: $\left\{{\left(PS\right)}_{1},{\left(PS\right)}_{2},{\left(PS\right)}_{3}\right\}\in \mathrm{MLB},\;\left\{{\left(PS\right)}_{5},{\left(PS\right)}_{6},{\left(PS\right)}_{7}\right\}\in \mathrm{ORP},\left\{{\left(PS\right)}_{9},{\left(PS\right)}_{10},{\left(PS\right)}_{11}\right\}\in \mathrm{OPP}$. Moreover, for each PS type, we supposed that there would be e number of parking slots ${\left(PL\right)}_{e}$, where PL denotes a parking slot, such that:(2)$${\left(PL\right)}_{e}=\left\{{\left(PL\right)}_{1},{\left(PL\right)}_{2},\ldots \ldots ,\;{\left(PL\right)}_{e-1},{\left(PL\right)}_{e}\right\}\;\;$$

However, depending on the size of each PS, the number of PL is fixed, such that ${\left(PL\right)}_{e-1}\leq \;{\left(PL\right)}_{e}\in \;{\left(PS\right)}_{n}$, where e and n are some fixed values. Each PL is installed with an RFID reader of the active type and is considered an edge node (EN) in the SAVP system. For indoor parking, an RFID reader was installed exactly in the middle of a PL, and each PL was connected to the fog node (FN) via routers installed to route the information between the PL and fog node. In the case of outdoor parking, each PL was also installed with an RFID reader, which could be mounted onto a panel or the pedestrian walkway, and that was assumed to be protected from any atmospheric obstacles. For both types of parking, we assumed that there were no obstacles and/or interference among the installed ${\left(PL\right)}_{e}$. In a PS, the ${\left(PL\right)}_{e}$ are connected to the ${\left(FN\right)}_{t,\;t=t-1}$, where t is some fixed value. Each FN is superior at managing the overall parking traffic in a PS and is connected to a cloud server (CS), such that a number of cloud servers are represented by ${\left(CS\right)}_{i,\;i=i-1},\;$ where i is some fixed value. A two-way communication between the $\;{\left(FN\right)}_{t,\;t=t-1}$ and ${\left(CS\right)}_{i,\;i=i-1}$ is carried out through GSM–LTE connectivity. Figure 4 illustrates a number of nodes that were assumed to have anonymous identifications and that participated in carrying the information in the SAVP system. From Figure 4, we can see that there is an FN (e.g., ${\left(FN\right)}_{1}$) that contains several edge nodes ${\left(EN\right)}_{z,\;z=z-1}$, where z is some fixed value, such that ${\left(EN\right)}_{z,\;z=z-1}\in {\left(FN\right)}_{1}\in$ ${\left(PS\right)}_{1}$. Therefore, in a PS (e.g., ${\left(PS\right)}_{1}$), there is only one ${\left(FN\right)}_{1}$ connected to a CS. As mentioned, each FN is fully functional to manage the overall parking operations of a PS; unlike the ${\left(EN\right)}_{z,\;z=z-1}$, the ${\left(FN\right)}_{t,\;t=t-1}$ are supposed to be installed with enough computation and energy resources to manage the flow of massive traffic in a PS (for example, from the AV entrance to exit). Likewise, each FN is also authorized to manage the overall operations and measurements from the ${\left(EN\right)}_{z,\;z=z-1}\;$ in a PS, so that each FN can manage the operations of the CS closer to the ${\left(EN\right)}_{z,\;z=z-1}$. As per the defined rules, an FN can only share the information with the designated CS (for example, ${\left(FN\right)}_{1}↔{\left(CS\right)}_{1}$). Likewise, ${\left(EN\right)}_{z,\;z=z-1}↔{\left(FN\right)}_{1}↔{\left(CS\right)}_{1}$.

In Figure 5, an FN is responsible for performing the security checks in order to permit an AV into parking, manage and store a massive number of transactions from the defined ${\left(EN\right)}_{z,\;z=z-1}$, and record the transactions onto the blockchain. To limit the storage onto the ledger, we only recorded the hash (H) of each transaction (T) that was further chained after the successful validation through the proof of transaction. This means that each carried transaction (T) from each EN is stored in the local storage, and only the hash (H) of each T (H(T)) is recorded each time onto the blockchain ledger of an FN. A transaction (T) could be of many types: an AV is authenticated, and is then authorized to enter a PS, the AV uses a PL, and finally, the AV exits the PS; therefore, each operation performed (e.g., from parking entrance to exit) is considered a transaction (T). To carry out a transaction (T) at the PS entrance, each AV is identified via the active tag that is embedded within it, and to ensure privacy, each tag has a unique anonymous identification (ID), one-time usable session key ($K_{ss}$), encrypted ID ($E({\left(ID\right)}_{K_{ss}})),\;$and a hash of IDs ($H\left(ID\right)),\;\mathrm{such}\;\mathrm{that}\;\mathrm{AV}\left\{\left(ID\right),K_{ss},\;E({\left(ID\right)}_{K_{ss}}),H\left(ID\right)\;\right\}$; therefore, upon PS entrance, an AV transmits its $ID,\;E({\left(ID\right)}_{K_{ss}}$, and H(ID), which are considered the attributes of the AV, and an FN will verify the AV’s attributes by employing the LIBC module to obtain access to the PS. As the $K_{ss}$ is of the symmetric type, in the first stage, the FN uses its $K_{ss}$
*(*$FN\left(K_{ss}\right)),$ which is associated with an AV, to perform the decryption (D): $FN\left(K_{ss}\right):\;D\left\{\;E({\left(ID\right)}_{K_{ss}})\;\right\}$. Both the encryption (E) and decryption (D) operations use a unique session key ($K_{ss}$) so that, at this stage, the AV can be authenticated only if $\mathrm{AV}\left(K_{ss}\right)=FN\left(K_{ss}\right)$. In the second stage, the integrity of the AV’s ID is verified: the FN computes the hash (H) of the received AV’s IDs (FN: H(ID)), and then compares the received hash (H) of the AV, such that FN: H(ID)= AV: H(ID). Thus, if both the hashes match, the FN can verify that the ID of the AV has not been altered. Upon successful decryption (D) and hash (H) verification, the AV will be allowed to access the PS. Similarly, each FN keeps the information of that AV parking on a designated PL, and finally, upon exit from the PS.

As per the requirements, there are four main communication scenarios that are employed to offer reliable two-way communication and manage the flow of information in the SAVP system: edge-to-fog, fog-to-cloud, fog-to-fog, and cloud-to-cloud. In the edge-to-fog scenario, the ${\left(EN\right)}_{z,\;z=z-1}$ read the information that can be further shared only with the designated FN in a PS; therefore, the ${\left(EN\right)}_{z,\;z=z-1}\;$ are entirely dependent on FN instructions to execute the desired operations. In the fog-to-cloud scenario, the overall operations of a PS are locally managed and controlled by an FN, which is an efficient intermediate node that runs the services of a CS to make the parking activities operational. Each FN in the PS was installed and configured in a way to manage the overall PS operations, without any limitations in terms of computing, storage, and power resources. However, in the long run, each FN is dependent on a CS; for example, there is a massive amount of data that may be computed from the ${\left(EN\right)}_{z,\;z=z-1}$ and that further need to be stored as the permanent storage that is backend CS storage. Thus, after the passage of time, the computed and/or recorded data onto an FN will be transferred to the CS. Among others, the CS performs the usual monitoring services to maintain the QoS on the FN, which also includes updating the FN with new information. For example, in a situation where a new AV is being registered onto the cloud in the SAVP system, the CS will also update the record onto the FN. The fog-to-fog scenario is a special communication scenario in which the ${\left(FN\right)}_{t,\;t=t-1}$ from several distinct ${\left(PS\right)}_{n,n=n-1}$ are grouped in hierarchical order to manage, store, and share information. The ${\left(FN\right)}_{t,\;t=t-1}$ are grouped into a chain to offer reliable communication, and to ensure a proper flow of information and QoS, as per the requirements of AVs. The fog-to-fog scenario plays an important role in the exchange of information with other ${\left(FN\right)}_{t,\;t=t-1}$ that may belong to distinct ${\left(PS\right)}_{n,n=n-1}$, as all the ${\left(FN\right)}_{t,\;t=t-1}$ are running similar services and employing resources from the backend cloud servers (${\left(CS\right)}_{i,\;i=i-1}$). The most important uses for this scenario are to perform a proof of transaction and to record each transaction onto the neighboring ${\left(FN\right)}_{t,\;t=t-1}$. For example, upon the successful confirmation of parking as per the request of an AV, the information as a successful transaction (T) (e.g., upon entrance or exit) is computed and recorded onto the ${\left(FN\right)}_{t,\;t=t-1}\;$, which are authorized participating nodes in the SAVP system. Upon the validation of each transaction, all participating ${\left(FN\right)}_{t,\;t=t-1}$ are provided with a replicated record to ensure that the AV was able to obtain parking service in the PS, and that there was no other authorized parking request or parking with a similar ID. In the cloud-to-cloud scenario, with the important checks onto the ${\left(FN\right)}_{t,\;t=t-1}$ to ensure QoS, the ${\left(CS\right)}_{i,\;i=i-1}$ participated to perform the proof of transaction to: (1) doublecheck that the data were synchronized and recorded onto the ${\left(FN\right)}_{t,\;t=t-1}$ and then uploaded onto the ${\left(CS\right)}_{i,\;i=i-1}$; (2) ensure that all the ${\left(CS\right)}_{i,\;i=i-1}$ were keeping an exact replicated record from the ${\left(FN\right)}_{t,\;t=t-1}$. This scenario is highly amenable to running a scalable SAVP system, as it has hundreds of millions of ${\left(EN\right)}_{z,\;z=z-1}$ connected with ${\left(FN\right)}_{t,\;t=t-1}$; however, this scenario is feasible if and only if, together, several ${\left(CS\right)}_{i,\;i=i-1}$ from different vehicular companies are running on a collaborative platform to manage parking services and perform careful checks on the performances of their vehicles while on the road.

## 5. Results and Discussion

This study aimed to conduct a proof-of-concept implementation for a SAVP system to manage the parking services for AVs. For that, a SAVP system was designed and implemented in C#, the Azure cloud platform was used for the virtualization of the edge and fog nodes, and the Azure SQL database was used to manage and run the blockchain services on the cloud servers, as well as on the fog nodes. Due to resource constraints and service costs, blockchain platforms such as Ethereum or Hyperledger Fabric were not within the scope of this study. Overall, the results were simulated and conducted on a personal computer running with a 64-bit Windows 10 Professional operating system (OS) and installed with 16 GB of memory (RAM) (Intel^®^ Core ™ i7-8665U CPU @ 1.90 GHz 2.11 GHz).

To compute the results, we limited the number of ${\left(EN\right)}_{z,\;z=z-1}$ in the ${\left(PS\right)}_{n,n=n-1}$, such that:$$\mathrm{MLB}:\;\left\{{\left(EN\right)}_{1},{\left(EN\right)}_{2},{\left(EN\right)}_{3}\right\}\in {\left(FN\right)}_{1}\in {\left(CS\right)}_{1};$$
$$\mathrm{ORP}:\;\left\{{\left(EN\right)}_{5},{\left(EN\right)}_{6},{\left(EN\right)}_{7}\right\}\in {\left(FN\right)}_{2}\in {\left(CS\right)}_{2};$$
$$\mathrm{OPP}:\;\left\{{\left(EN\right)}_{9},{\left(EN\right)}_{10},{\left(EN\right)}_{11}\right\}\in {\left(FN\right)}_{3}\in {\left(CS\right)}_{3}$$

Each PS was installed with an intermedia FN. In the same way, each FN was connected to a backend CS (a virtual server, in our case) to deploy the cloud services or operations closer to the ${\left(EN\right)}_{z,\;z=z-1}$, and to share the carried information with the designated CS. As mentioned, the SAVP system only offers parking services to registered AVs. For simplicity, let us suppose that an AV is from Tesla, which is a vehicular company that aims to start having functional AVs on the road for services such as self-driving taxis; therefore, the AV needs to register under the SAVP system to obtain the parking service of any PS, either indoors or outdoors, under the regulations of the SAVP system. To perform the AV authentication and verification as parts of the LIBC module, the given conditions, such as $\mathrm{FN}\left(K_{ss}\right)=\mathrm{AV}\left(K_{ss}\right)$ and FN: H(ID)= AV: H(ID), need to be satisfied in Stages 1 and 2. To ensure privacy in each scenario, we assumed that each participating node had a unique anonymous identification to communicate in the SAVP system. Figure 6, Figure 7 and Figure 8 illustrate a selective number of measurements that are computed in MLB, ORP, and OPP under the edge-to-fog scenario; a number of time experiments were performed to enable reliable access to AVs looking for parking. Figure 6 shows the results computed in 10 min; likewise, Figure 7 and Figure 8 display the results that were measured in periods of 20 min and 30 min, respectively. Notably, the $K_{ss}$ always expires within 2 s (maximum); thus, in order to measure a successful transaction (T), a transaction should be performed within 2 s. Therefore, from Figure 6, Figure 7 and Figure 8, we believe that an AV that is being registered and obtaining access as per the satisfaction of authentication and verification at Stages 1 and 2 and/or as per the limits of the $K_{ss}$ will be counted as an authorized AV to the SAVP system. In the case that an AV is not successful in authenticating its identity via the $K_{ss}$ and verification through hash (H), the SAVP system will then deny access to that AV. However, each AV has three chances to prove its identity at the entrance of a PS; otherwise, the $K_{ss}$ will be invalid so that the AV will need to manage or request a new $K_{ss}\;$ from the CS, which was the only entity authorized to issue a new key in our case study. Thus, the CS is also responsible for performing the operations of a CA and KDC.

To achieve the desired study goals, a number of experiments were conducted in a situation that bypassed the FN operations so that the ${\left(EN\right)}_{z,\;z=z-1}$ were directly connected to the CS. To perform this, careful checks on the performance latency and efficiency of the SAVP system were kept in mind. Figure 9 demonstrates the results that were computed during the successful exchange of information from the ${\left(EN\right)}_{z,\;z=z-1}\;$ to the CS (for example, AV identification over the parking entrance without having enrollment in the FN). The selective measurements of Figure 9 show that the computed latency is almost twice the results computed in the edge-to-fog scenario. However, similar to other computed performances, the results of Figure 9 were also measured in the absence of any network obstacles or communication errors.

In the SAVP system, each successful transaction (T) is recorded onto the blockchain, and the proof of transaction is performed over the neighboring fog nodes: ${\left(FN\right)}_{t,\;t=t-1}$. Let us suppose that the parking stations, or ${\left(PS\right)}_{n,n=n-1}$, are regulated under the control of the SAVP system so that the SAVP system becomes a collaborative decentralized platform, where fog-to-fog and cloud-to-cloud scenarios operate together to share various resources, as well as to keep copies of all the transactions that are performed as part of the SAVP system. To offer a reliable parking solution, there are always three main transactions: AV$\left(T_{1}\right)$, AV$\left(T_{2}\right)$, and AV$\left(T_{3}\right)$, which are usually executed in a cascading manner: AV$(T_{1})\to$ AV$(T_{2})\;\to$ AV$\left(T_{3}\right)$. AV$\left(T_{1}\right)$ is performed upon entrance to the PS via the LIBC module, AV$(T_{2})\;$ is performed on the designated PL, and AV$\left(T_{3}\right)$ is fulfilled at the parking exit. These transactions are recorded first onto the FN or ${\left(FN\right)}_{1}$ locally, and are then recorded onto other fog nodes, such as ${\left(FN\right)}_{2},\;{\left(FN\right)}_{3}$, and so on, as the successful operations of the proof of transaction. By doing so, all fog nodes, or ${\left(FN\right)}_{t,\;t=t-1}$, keep an updated record of each AV that requested parking in any parking stations, or ${\left(PS\right)}_{n,n=n-1}$. Thus, the fog nodes are always updated with new information that may be carried on any fog node that is associated with a distinct cloud. Likewise, cloud servers obtain updated information via their associated fog nodes’ time-to-time and can perform the validation or the proof of transaction through the confirmation that all the clouds have replicated data carried from their associated fog nodes.

To minimize the storage space in an FN, each transaction (T) that carries data (for example, some random bits) is first recorded onto the FN and is then transferred to the CS after a specific time session, whereas the hash (H) of each transaction (T), or H(T), is only chained and replicated to other neighbor fog nodes. In this way, an FN can minimize its storage. Table 1 depicts a number of successful transactions ($T_{x}$), where x is a fixed value that is recorded onto the fog nodes, and then onto cloud servers. Let us suppose that ${\left(FN\right)}_{1}\;$ performed the transactions ($T_{x}$) so that successful $T_{x}\approx 1\;$are first recorded onto ${\left(FN\right)}_{1}$, and then onto neighboring fog nodes (i.e., ${\left(FN\right)}_{2},\;{\left(FN\right)}_{3},\ldots \ldots ,{\left(FN\right)}_{t-1}$). This situation is similar to recording transactions ($T_{x}$) onto cloud servers. Figure 10 summarizes the overall performance results to evaluate the efficiency of the proposed system. We can see that, in each scenario, edge-to-fog, fog-to-cloud, fog-to-fog, and cloud-to-cloud communication was initiated and successfully carried out by applying a session key of 2 s. However, the performance results as per each scenario are varied, but they are still under the limits of the session key. For communication scenarios such as edge-to-fog and fog-to-fog, all the associate responses from clouds are observed to evaluate the status of each cloud (either active or inactive). To the best of our knowledge, for each transaction (T), each cloud status was observed as an active status; therefore, from Figure 11, we can assume that each cloud response was always linear, or in order. Therefore, as per the study design and its overall measured results in each communication, we can conclude that this study is an extension of the existing available works [12,16,17,33,36], and of others that employ the emerging fog and blockchain technologies as add-ons to manage overall parking operations and security for the existing IoT–cloud-based SP systems [5,7,9,34]. In reality, IoT–cloud systems come with the heterogeneity of resource constraint that necessitates having a friendly and secure architecture to enable an efficient SP system.

## 6. Challenges and Future Works

The main challenges of the current study, as well as for the autonomous vehicular world, which we examined in depth and which will be targeted in the near future, are as follows:i.As several well-known vehicular companies are racing towards producing AVs [54], it is very interesting to have a smart-parking application installed in and supported with autonomous-vehicle built-in software; for example, the robot operating system (ROS), which is a flexible platform or tool to support and control AVs [55], and which can perform the functions of parking whenever is required by AVs. For example, whenever AVs are in an idle state, or not available for any service, the SP application can search for a closer available parking spot (e.g., either indoors or outdoors), and the interesting part is its electric charging services. The autonomous vehicle could obtain the parking spot and fulfill the demands of electric charging if required by the autonomous vehicle, as most AVs are supposed to be electric. However, if, for example, the autonomous vehicle has enough electric charging and thus parks in a parking spot that has no electric charging capabilities, then parking spots will be available for other AVs that have limitations to their electric charging services. In the future, there will be high demand for electric vehicles globally; as the report states, almost 130,000 electric vehicles were sold out in 2021, with about 6.6 million sold [56];ii.In metropolitan cities around the globe, multilevel parking is one of the best ways to offer and manage a large number of vehicles that are searching for parking, as per the indented closer locations. However, in the case of AVs, it is a challenging problem to offer indoor or multilevel parking, as the GPS is not functional inside the premises of multilevel buildings [57,58]. To address this major issue, indoor driving technology, including indoor positioning systems (IPS) and smart sensor technology, can be employed as a resilient solution to conquer the challenge of indoor multilevel parking [59,60,61,62,63,64]. Moreover, in large cities, where there are supposed to be numerous parking stations, these designs, or maps, can be varied as per the indoor or multilevel architectures, as it is a challenging issue for AVs to follow distinct parking designs to perform accurate parking in short periods of time. Among other things, in various essential parking designs, each autonomous vehicle should have installed, for example, an individual IPS for each parking station that functions on the road, or an autonomous vehicle design (i.e., firmware) that is fully functional or/and has capabilities to auto-install the IPS, or indoor driving map, of the parking station before entering; however, the best solution is that AVs should be installed with the IPS of each parking station, or with the IPSs of the parking stations that are essential for use in daily service routines; therefore, by having this feature, the autonomous vehicle can obtain information on closer parking stations, with the availability of parking slots;iii.The exponential developments of 5G, the IoT, M2M, and other modern technologies can play important roles in enabling fully autonomous vehicles on the road with entirely safe features; however, having millions of AVs on the road will come with several challenges, which include, but are not limited to, the interoperability for both hardware and software. There are several vehicular companies that have already started or that are in the process of building fully AVs, with distinct vendors and vehicular companies. AVs on the road need QoS and intelligent information sharing for the safety of humans, and also for the safety of the AV itself. Therefore, there will be a requirement to build a collaborative generic platform, while keeping the overall interoperability issues of hardware and software in mind, to enable reliable communication and information sharing among millions of AVs while on the road, as the platform is collaborative; thereby, vehicular companies will be able to monitor the overall performances of their vehicles on the road in real time. However, depending on the polices and defined rules, a vehicular company can be restricted in the specific information that it can access and share with others;iv.There are several cloud and blockchain platforms, such as public, private, and hybrid, and their services have been offered by numerous companies, such as IBM, Google, and Oracle, in addition to others. Similar to other sectors, vehicular companies will install their AVs with distinct cloud and blockchain platforms; thus, there will the requirement to have a collaborative platform to share information among different clouds and blockchain platforms. For example, let us suppose that there is a vehicular company (A) that employs Google Cloud and the blockchain service by Ethereum, and there is another company (B) that installs the IBM cloud and the Hyperledger Fabric platform. There is a necessity to have a collaborative platform to share information without any limits of interoperability, availability, security, etc. This platform is important, but it is a challenging problem to have AVs from several companies on the road;v.In the near future, there will probably only be AVs on the road; for this, an intelligent platform will be required to monitor and manage a huge number of resources and information 24 h a day, or whenever the AVs function to perform any service, with a guarantee of maintaining better QoS. To offer better QoS, a more resilient hybrid cloud platform will be required to manage a large amount of information and resources. Offering a hybrid cloud service will be a better solution, as the vehicular companies will need to participate in collaboration to run their millions of vehicles on the road. Without collaboration, it will probably be impossible to operate AVs from different companies on the road. However, there should be a cybersecurity mechanism that employs cryptography, blockchain, etc., to ensure the overall information between vehicular companies having either private or hybrid cloud servers, hundreds of millions of AVs, parking stations, etc. For this, and particularly to manage such a crowded AV infrastructure with concrete reliable cybersecurity mechanisms, introducing distributive cloud-based blockchain approaches will be a better solution to combatting the challenges and to ensuring secure two-way communication in an autonomous world.

## 7. Conclusions

This study makes an interesting and important contribution to the arena of smart parking. To address the current and future challenges of scalable parking, wherein hundreds of millions of vehicles will be on the road and searching for parking, this study employed emerging fog–blockchain technologies to strengthen the existing IoT–cloud platform for AVs. Looking towards the future, where there will only be AVs on the road, and they will look for parking. It is a challenging issue to enable an SP solution for such a massive number of autonomous vehicles to offer QoS at each stage of parking and to deliver the best performances, such as a minimal latency, system efficiency, and security. For this, this study deployed an inclusive, long-term, effective, and well-performing SAVP system to address the challenges of the current and future SP systems, and to enable an SP solution, both for indoor and outdoor parking, for AVs. A proof-of-concept implementation was conducted for an SAVP system, where the performances were measured in the presence of four communication scenarios: edge-to-fog, fog-to-cloud, fog-to-fog, and cloud-to-cloud. The introduction of fog nodes into the existing IoT–cloud platform could have a positive impact in terms of throughput and efficiency, as the operations are performed closer to the edge nodes. Introducing an LIBC module into the fog level is a better solution to combat the security challenges of authentication and verification, as this grants access to authorized AVs only. Every successful transaction that is carried from the edge nodes, or that is based on the communication scenarios, is recorded onto the blockchain after a validation process called the proof of transaction. Based on the performance results, we believe that the proposed SAVP system is highly efficient at reducing the average throughput, boosting the system efficiency, and, importantly, addressing the issues of security and privacy, and we examined it as an innovative pace to manage future SP systems for AVs.

## Figures and Tables

**Figure 1 sensors-22-04849-f001:**
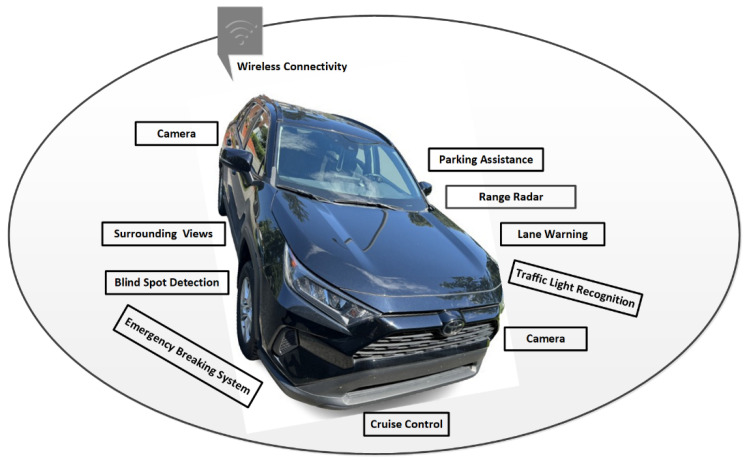
Modern Vehicular Technologies.

**Figure 2 sensors-22-04849-f002:**
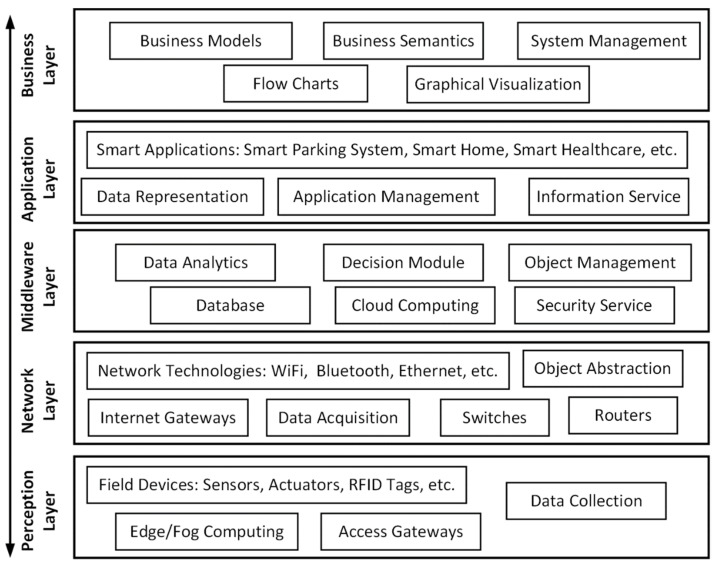
IoT five-layer architecture.

**Figure 3 sensors-22-04849-f003:**
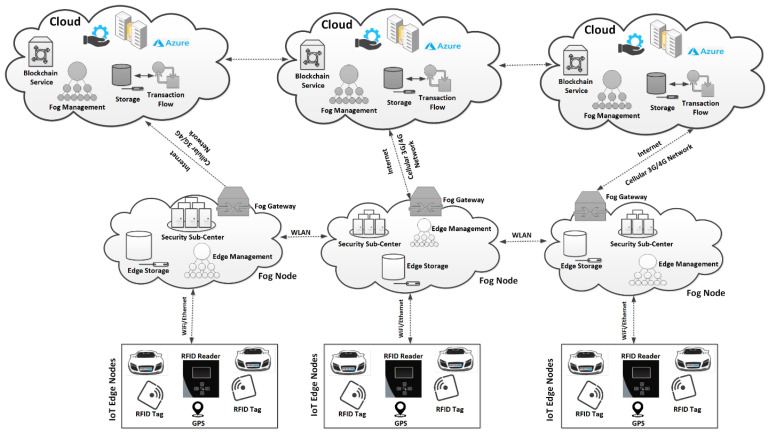
System Architecture.

**Figure 4 sensors-22-04849-f004:**
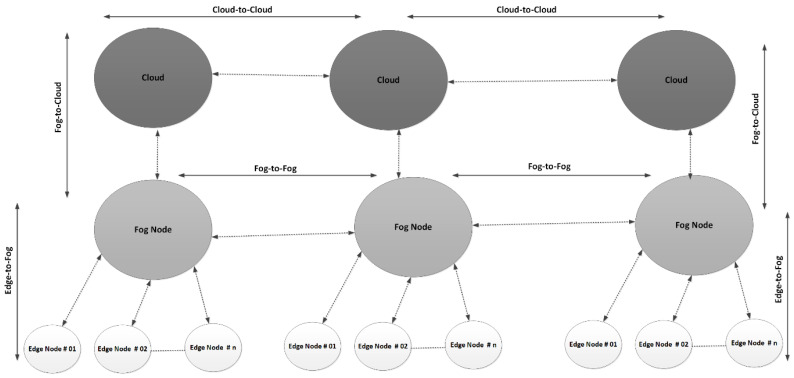
Node Configuration and Setup.

**Figure 5 sensors-22-04849-f005:**
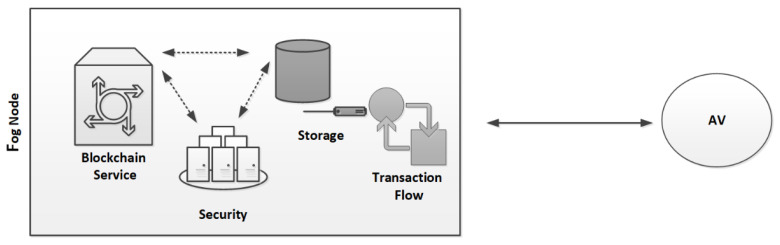
LIBC Module.

**Figure 6 sensors-22-04849-f006:**
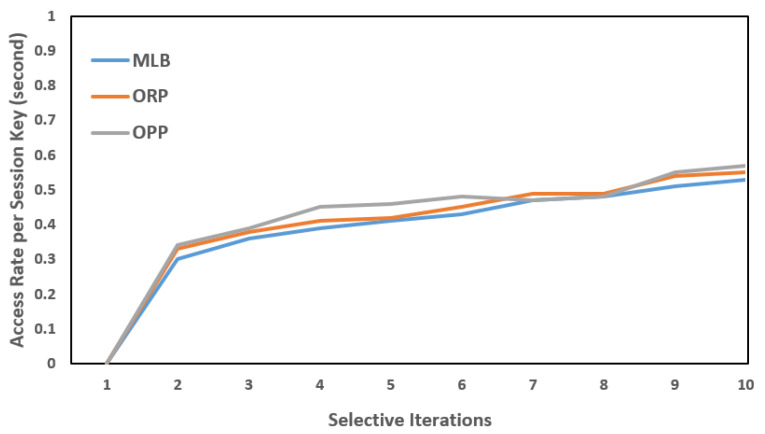
AV access rates per session key: key session = 2 s; time period = 10 min.

**Figure 7 sensors-22-04849-f007:**
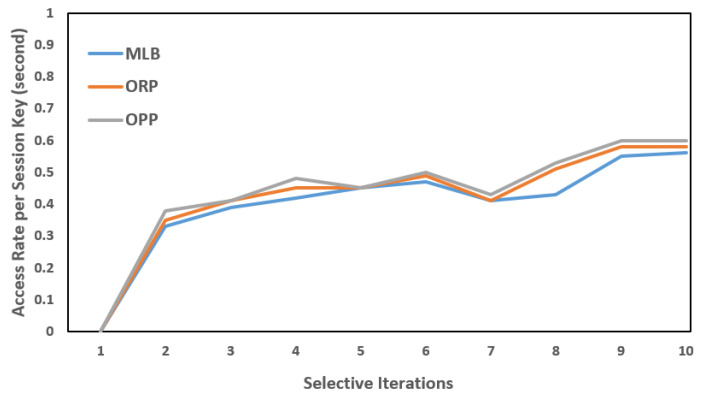
AV access rates per session key: session key = 2 s; time period = 20 min.

**Figure 8 sensors-22-04849-f008:**
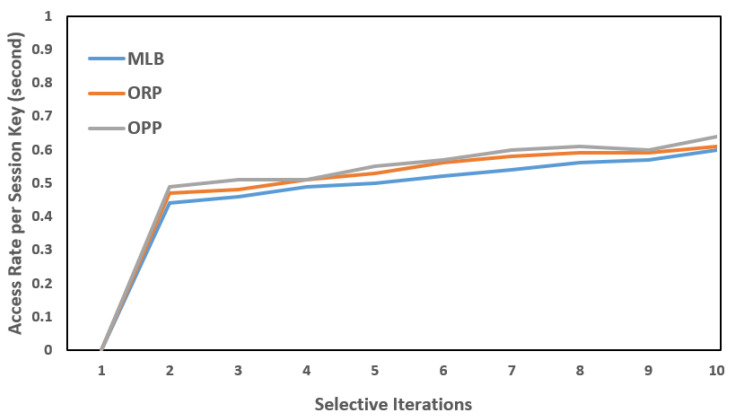
AV access rates per session key: session key = 2 s; time period = 30 min.

**Figure 9 sensors-22-04849-f009:**
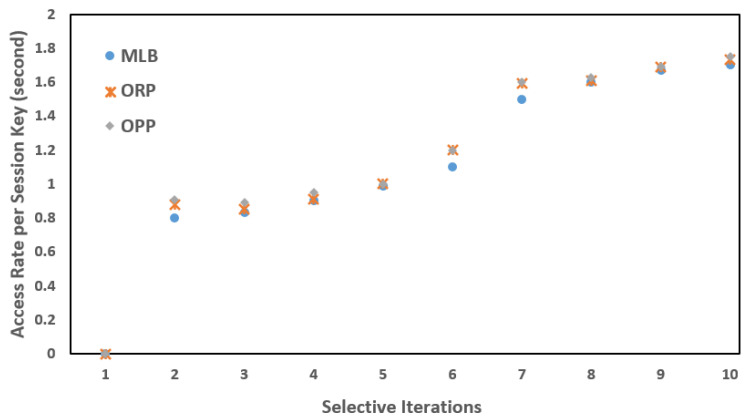
AV access rates per session key via the cloud: session key = 2 s; time periods: 10, 20, and 30 min.

**Figure 10 sensors-22-04849-f010:**
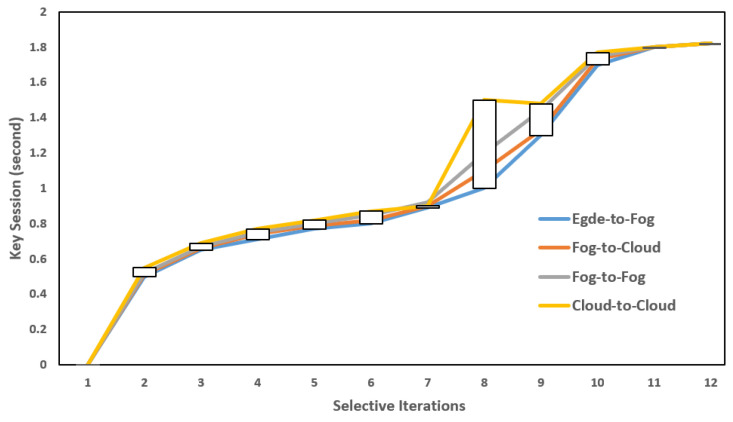
Performance results per communication scenario: session key = 2 s; time periods: 10, 20, and 30 min.

**Figure 11 sensors-22-04849-f011:**
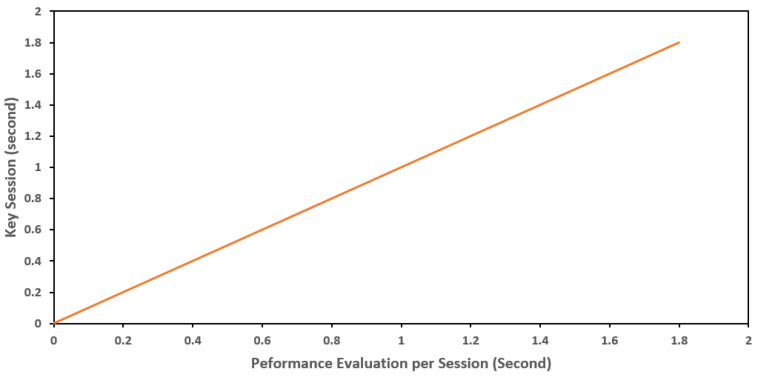
Performance Evaluation.

**Table 1 sensors-22-04849-t001:** Proof of Transaction.

Scenario	H, 256 Bits	FN ; CS		
		${\left(FN\right)}_{t=1}$ $;\;{\left(CS\right)}_{i=1}$	${\left(FN\right)}_{t=2};$ ${\left(CS\right)}_{i=1}$	${\left(FN\right)}_{t=t-1}$ $;\;{\left(CS\right)}_{i=i-1}$
Fog-to-Fog	ba8730e3516cf4d509419689446fe04fd564d0a373a185772b8d4d4d95c8b73e	≈1	≈1	≈1
	d843c2fc386f12ab0efe8e54b8e34e8a1e55e58cc98dc58bbd0e35c28f0597aa	≈1	≈1	≈1
	8cd71644c06b38d4476adcf3b4095dbf3d5ef7bbaf3e6a511c127e642a39017d	≈1	≈1	≈1
	096ec92882f3dfba23b364431d3d7cc728aa9ace60ca47d176645298eaf05e1d	≈1	≈1	≈1
	d63fd9d273859a84a0a94580de7341f3544c1ee76e790acc715c1039c1bf9d54	≈1	≈1	≈1
Cloud-to-Cloud	14fb562fa8e9915afc5dee2bc45613e37f4cb9f6fdaee556df7484d08c9879c6	≈1	≈1	≈1
	3df5057469639c1f983ed32442db32c139e10385b4a0f087291869e3e7aa5a30	≈1	≈1	≈1
	fab490c50246accfc538bd1ed5e47566980c2d4244721ea6b0383ab3a9ab1780	≈1	≈1	≈1
	505a490dbca80a72660a9983abb7d2b5b6ceab939b80dc44dec29272879ae511	≈1	≈1	≈1
	bbcd939cc5e48ac1d4b5581ba142623477ddf9b22df4c9e8ad5e4d4f9c0182d8	≈1	≈1	≈1

## Data Availability

Data sharing is not applicable.

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
