# Peer review of "Enabling Fog–Blockchain Computing for Autonomous-Vehicle-Parking System: A Solution to Reinforce IoT–Cloud Platform for Future Smart Parking"

_sensors, 2022, doi:10.3390/s22134849_

Round 1
Reviewer 1 Report
The manuscript deals with a smart autonomous parking system of AVs to address the various issues and challenges. A proof-of-concept implementation is well-presented and provides realistic communication scenarios.
The topic is interesting and relevant to the journal, but there are major issues that prevent publication.
- The manuscript is poorly formatted; resubmission is required.
- For Figure 1, copyright must be checked.
- Introduction is too general and fails to deliver research background and motivation.
- The manuscript outline is wrongly written (last part of Introduction).
- Scientific representation is required especially for equations.
- No pseudocodes or algorithms are provided.
- For Figure 10, the label of x-axis can be revised.
- I recommend moving Section 8 before the conclusion section as discussion.
- There are many typos and errors. I recommend consulting a manuscript editing service.
- More recent references are required.
- Comparative analysis with the state of the art studies are required.
Overall. the manuscript incomplete and requires a re-submission after major issues are resolved.
Author Response
Reviewer 1 Comments/Suggestions and Authors Feedback.
|
Comment/Suggestion 01: Extensive editing of the English language and style required. Feedback 01: Overall manuscript has been thoroughly checked for English language corrections. |
|
Comment/Suggestion 02: The manuscript is poorly formatted; resubmission is required. Feedback 02: Manuscript has been formatted in an order. |
|
Comment/Suggestion 03: For Figure 1, copyright must be checked. Feedback 03: Actually, the vehicle illustrated in Figure 1 is my vehicle and the given text is rearranged; therefore, we don’t need any copyright. |
|
Comment/Suggestion 04: Introduction is too general and fails to deliver research background and motivation. Feedback 04: Please see the last three paragraphs of Section 1, to have an interesting research background and potential motivations for this study. |
|
Comment/Suggestion 05: The manuscript outline is wrongly written (last part of Introduction) Feedback 05: The last part of Section 1 has been revised for corrections. |
|
Comment/Suggestion 06: Scientific representation is required especially for equations. Feedback 06: The mathematical equations of the manuscript have been revised. |
|
Comment/Suggestion 07: No pseudocodes or algorithms are provided. Feedback 07: We believe, there is no requirement for pseudocode or algorithm. However, the new information is added in sections 3 and 4, to make a precision. |
|
Comment/Suggestion 08: For Figure 10, the label of x-axis can be revised. Feedback 08: Figure 10 has been revised. |
|
Comment/Suggestion 09: I recommend moving Section 8 before the conclusion section as a discussion. Feedback 09: Section 8 has been switched before the conclusion. |
|
Comment/Suggestion 10: There are many typos and errors. I recommend consulting a manuscript editing service. Feedback 10: Overall manuscript has been revised for corrections such as typos ad errors. |
|
Comment/Suggestion 11: More recent references are required. Feedback 11: References such as 1, 58, 61, and others are somehow recent and close relative to this study. |
|
Comment/Suggestion 12: Comparative analysis with the state-of-the-art studies is required Feedback 12: More useful information is added in Sections 4 and 5, to demonstrate the significance of the proposed study compared to the existing studies. |
|
Comment/Suggestion 13: Overall. the manuscript is incomplete and requires a re-submission after major issues are resolved. Feedback 13: Overall manuscript has been revised. Therefore, we wish that the revised version of the manuscript will fulfill your requirements.
|

Reviewer 2 Report
Research work is very interesting and the article is well written.
Some points that need to address are as follows.
Change the given name to the proposed system as it conflicts with the German multinational software corporation.
in introduction, the contribution given in points, make them short nd precise and move the description before or after the numbered list.
MLB, ORP, and OPP is good analysis but should compare your proposed system with any of the published related work [11], [17], [36], [33], or other but shall be from high impact journal to justify your work in a strong manner.
enhance abstract, introduction, and conclusion with new comparisons and analysis.
Consider other most relevant high impact journals
MDPI Sensors: 14 articles >> https://www.mdpi.com/search?q=iot+smart+parking&journal=sensors
IEEE Transactions on Vehicular Technology
IEEE TRANSACTIONS ON INTELLIGENT TRANSPORTATION SYSTEMS
ieee transactions industrial informatics
IEEE Internet of Things Journal
IEEE Communications Magazine
87 for IoT Parking find with this query (https://ieeexplore.ieee.org/search/searchresult.jsp?queryText=IoT%20Parking&highlight=true&returnType=SEARCH&matchPubs=true&refinements=ContentType:Journals&refinements=PublicationTitle:IEEE%20Internet%20of%20Things%20Journal&refinements=PublicationTitle:IEEE%20Transactions%20on%20Intelligent%20Transportation%20Systems&refinements=PublicationTitle:IEEE%20Transactions%20on%20Vehicular%20Technology&refinements=PublicationTitle:IEEE%20Sensors%20Journal&refinements=PublicationTitle:IEEE%20Transactions%20on%20Industrial%20Informatics&refinements=PublicationTitle:IEEE%20Transactions%20on%20Automation%20Science%20and%20Engineering&refinements=PublicationTitle:IEEE%20Communications%20Surveys%20.AND.%20Tutorials&ranges=2018_2022_Year&returnFacets=ALL),
you can narrow down these 87 by including blockchain and fog computing keywords.
Search from other high impact journals as follows and consider
Computer Communications
Sustainable Cities and Society
also update literature review with the important relevant published work. Show clear strength of your work in front of the published work, (better with the help of comparison table)
Author Response
Reviewer 2 Comments/Suggestions and Authors Feedback.
|
Comment/Suggestion 01: English language Corrections. Feedback 01: Overall manuscript has been thoroughly checked for English language corrections. |
|
Comment/Suggestion 02: Research work is very interesting, and the article is well written. Feedback 02: Thank you for your appreciation. |
|
Comment/Suggestion 03: Some points that need to address are as follows. Change the given name to the proposed system as it conflicts with the German multinational software corporation. Feedback 03: Over the world, several companies and systems have similar names, the important to have distinct abbreviations. So, in our case, the abbreviation of the proposed system is entirely different from others. |
|
Comment/Suggestion 04: In introduction, the contribution given in points, make them short and precise and move the description before or after the numbered list. Feedback 04: The contributions given in the introduction section have been revised. |
|
Comment/Suggestion 05: MLB, ORP, and OPP is good analysis but should compare your proposed system with any of the published related work [11], [17], [36], [33], or other but shall be from high impact journal to justify your work in a strong manner. Feedback 05: For that, more useful information is added in sections 3 and 4. |
|
Comment/Suggestion 06: Enhance abstract, introduction, and conclusion with new comparisons and analysis. Feedback 06: These sections have been revised for any further improvements. |
|
Comment/Suggestion 07: Consider other most relevant high impact journals. MDPI Sensors: 14 articles >> https://www.mdpi.com/search?q=iot+smart+parking&journal=sensors; IEEE Transactions on Vehicular Technology; IEEE TRANSACTIONS ON INTELLIGENT TRANSPORTATION SYSTEMS; ieee transactions industrial informatics; IEEE Internet of Things Journal; IEEE Communications Magazine; 87 for IoT Parking find with this query (https://ieeexplore.ieee.org/search/searchresult.jsp?queryText=IoT%20Parking&highlight=true&returnType=SEARCH&matchPubs=true&refinements=ContentType:Journals&refinements=PublicationTitle:IEEE%20Internet%20of%20Things%20Journal&refinements=PublicationTitle:IEEE%20Transactions%20on%20Intelligent%20Transportation%20Systems&refinements=PublicationTitle:IEEE%20Transactions%20on%20Vehicular%20Technology&refinements=PublicationTitle:IEEE%20Sensors%20Journal&refinements=PublicationTitle:IEEE%20Transactions%20on%20Industrial%20Informatics&refinements=PublicationTitle:IEEE%20Transactions%20on%20Automation%20Science%20and%20Engineering&refinements=PublicationTitle:IEEE%20Communications%20Surveys%20.AND.%20Tutorials&ranges=2018_2022_Year&returnFacets=ALL); you can narrow down these 87 by including blockchain and fog computing keywords; Search from other high impact journals as follows and consider; Computer Communications; Sustainable Cities and Society Feedback 07: More suitable citations are made in the manuscript. |
|
Comment/Suggestion 08: also update literature review with the important relevant published work. Show clear strength of your work in front of the published work, (better with the help of comparison table) Feedback 08: The literature review section of the manuscript has been checked, and we believe that the used published works are relevant to this study in many dimensions. |

Reviewer 3 Report
The Language of the manuscript-at-hand needs considerable improvement. Significant issues pertinent to Sentence Structure are apparent in the entire manuscript too.
Why there is a need for employing Blockchain for the envisaged mechanism as there are conventional security and trust-based techniques already in the literature. It is perfectly fine to be bias to a particular technique, however, it is only correct if you had taken other techniques, at least, into consideration and highlighted their inabilities to tackle the studied problem. Always built your stance in a more logical manner. Also, line 295, "the icing on the cake", is pretty absurd, i.e., this so-called "icing" is already out there and not done by the authors in this particular manuscript.
The notion of deploying RFID Tags on a vehicle is a very established one, and therefore, can't be claimed as one of the Contributions of this manuscript.
The Architecture presented in Figure 3 is standard and there is no novelty in the same. The System Model is also very basic and is presented in an abrupt manner. The Simulation Results are not convincing by any means and Critical Analysis of the same is missing here.
The crux of the entire manuscript is just delineated on 6 pages and most of the space has been utilized in the sections, Introduction, Literature Review, and Challenges and Future Works.
Author Response
Reviewer 3 Comments/Suggestions and Authors Feedback.
|
Comment/Suggestion 01: Extensive editing of English language and style required: The Language of the manuscript-at-hand needs considerable improvement. Significant issues pertinent to Sentence Structure are apparent in the entire manuscript too. Feedback 01: Overall manuscript has been thoroughly checked for English language corrections. |
|
Comment/Suggestion 02: Why there is a need for employing Blockchain for the envisaged mechanism as there are conventional security and trust-based techniques already in the literature. It is perfectly fine to be bias to a particular technique, however, it is only correct if you had taken other techniques, at least, into consideration and highlighted their inabilities to tackle the studied problem. Always built your stance in a more logical manner. Also, line 295, "the icing on the cake", is pretty absurd, i.e., this so-called "icing" is already out there and not done by the authors in this particular manuscript. Feedback 02: Thank you for such an interesting comment/suggestion. To do so, suitable corrections have been made in the appropriate sections of the manuscript. |
|
Comment/Suggestion 03: The notion of deploying RFID Tags on a vehicle is a very established one, and therefore, can't be claimed as one of the Contributions of this manuscript. Feedback 03: Yes, you are right! However, one of the targets of this study is to employ RFID, e.g., of the active type that is much more efficient to store a session key, hash of ID, and other security attributes. However, more suitable information is added to precise the study targets. |
|
Comment/Suggestion 04: The Architecture presented in Figure 3 is standard and there is no novelty in the same. The System Model is also very basic and is presented in an abrupt manner. The Simulation Results are not convincing by any means and Critical Analysis of the same is missing here. Feedback 04: In section 3, a piece of useful information with an additional figure is added to present a proper flow of information as per system architecture. To demonstrate the results, we revised the existing text and useful information has been added to show the potential of the study. |
|
Comment/Suggestion 05: The crux of the entire manuscript is just delineated on 6 pages and most of the space has been utilized in the sections, Introduction, Literature Review, and Challenges and Future Works. Feedback 05: More useful information is added in Sections 3, 4, and 5, to detail the proposed system architecture, SAP system model, and the computed results with the detailed discussions. |

Round 2
Reviewer 1 Report
The authors revised the manuscript based on the previous review, but the manuscript is poorly formatted.
Thus, I recommend the manuscript for publication after revising the manuscript format.
Author Response
Round 2: Reviewer 1 Comments/Suggestions and Authors Feedback.
|
Comment/Suggestion 01: The authors revised the manuscript based on the previous review, but the manuscript is poorly formatted. Thus, I recommend the manuscript for publication after revising the manuscript format. Feedback 01: Manuscript has been formatted in an order. |

Reviewer 2 Report
Comment regarding SAP not addressed
What is SAP and search SAP on google
https://www.sap.com/mena/about/company/what-is-sap.html
Comment/Suggestion 04: In introduction, the contribution given in points, make them short and precise and move the description before or after the numbered list.
Feedback 04: The contributions given in the introduction section have been revised
> But is still the same. <
Comment/Suggestion 05: MLB, ORP, and OPP is good analysis but should compare your proposed system with any of the published related work [11], [17], [36], [33], or other but shall be from high impact journal to justify your work in a strong manner.
Feedback 05: For that, more useful information is added in sections 3 and 4.
> Information that you added in section 3 and 4 is good but the comparison is required to justify the strength of the proposed system. As, multiple-level building (MLB), and the outdoor parking that includes on-road parking (ORP) and open premise parking (OPP) are scenarios. Shall compare with recent published work to show the betterment of your work in front of other available algorithms. <
Manuscript is well updated but left few important points.
Author Response
Round 2: Reviewer 2 Comments/Suggestions and Authors Feedback.
|
Comment/Suggestion 01: Comment regarding SAP not addressed. What is SAP and search SAP on google? https://www.sap.com/mena/about/company/what-is-sap.html Feedback 01: As per the respectable reviewer’s comment/suggestion, the SAP system has changed to the SAVP System. In short, an additional alphabet ‘V’ is added to make the difference. |
|
Comment/Suggestion 02(same as Round 1: Comment/Suggestion 04): In the introduction, the contribution given in points, make them short and precise, and move the description before or after the numbered list. Feedback 02: As per the respectable reviewer’s comment/suggestion, the contributions given in the introduction section have been revised. |
|
Comment/Suggestion 03 (same as Round 1: Comment/Suggestion 05): MLB, ORP, and OPP is good analysis but should compare your proposed system with any of the published related work [11], [17], [36], [33], or other but shall be from high impact journal to justify your work in a strong manner. Information that you added in section 3 and 4 is good but the comparison is required to justify the strength of the proposed system. As, multiple-level building (MLB), and the outdoor parking that includes on-road parking (ORP) and open premise parking (OPP) are scenarios. Shall compare with recent published work to show the betterment of your work in front of other available algorithms. Feedback 03: For that, a statement is added at the end of section 5: Therefore, as per the study design and its overall measured results in each communication, we can conclude that this study is an extension of the existing available works [12], [16], [17], [33], [36], and others, i.e., to employ the emerging fog and blockchain technologies as add-ons to manage overall parking operations and security for the existing IoT-cloud based SP systems [5], [7], [9], [34]. In reality, the IoT-cloud systems come with the heterogeneity of resources-constraint that necessitate having a friendly and secure architecture to enable an efficient SP system. |

Reviewer 3 Report
Thank you for incorporating the 'suggested changes'. The overall quality of the manuscript-at-hand has been considerably improved.
Author Response
Round 2: Reviewer 3 Comments/Suggestions and Authors Feedback.
|
Comment/Suggestion 01: Thank you for incorporating the 'suggested changes'. The overall quality of the manuscript-at-hand has been considerably improved. Feedback 01: Thank you for your appreciation. |
